# Doxazosin for the treatment of mental health disorders: A scoping review

**Christie S. Richardson**[1,2*], **Danah Atassi**[2], **Afshan Khan**[2], **Amanda Bernardini**[2], **Priyal P. Shah**[1]

**1** Department of Psychiatry, Rowan-Virtua School of Osteopathic Medicine, Stratford, New Jersey, United States of America, **2** Department of Psychiatry, Inspira Medical Center Vineland, Vineland, New Jersey, United States of America

* richardsonc@rowan.edu

## Abstract

Within the field of psychiatry, the alpha-1 adrenergic antagonist doxazosin is being studied for its off-label uses for disorders involving abnormal sympathetic nervous system arousal, including post-traumatic stress disorder (PTSD) and substance use disorders. Although prazosin is the most well-known alpha-1 antagonist used for these disorders, doxazosin has a longer half-life, which allows for more convenient once-daily dosing. Even though there is an advantage in dosing once daily for doxazosin, the medication is less well-known in the treatment of mental health disorders. The goal of this scoping review is to identify how doxazosin is currently being used in the treatment of mental health disorders, to determine if there is any evidence of its effectiveness for these disorders, and to identify promising areas for further research. PRISMA-ScR guidelines were used to identify articles describing doxazosin for the treatment of mental health disorders in PubMed, Embase, PsycINFO, and CENTRAL to understand current trends in research. Twenty-three articles met the inclusion criteria and included doxazosin for PTSD and/or nightmares, substance use disorders, and dual diagnoses. Many of the articles were small open-label trials or case reports. Existing evidence is strongest for doxazosin's use in PTSD-related nightmares and cocaine use disorder, which is consistent with the proposed alpha-1 antagonism effect on disorders with high levels of sympathetic nervous system arousal. Current clinical trials continue to be focused on PTSD and cocaine use disorder, given promising evidence. More randomized controlled trials need to be completed for recommendations of doxazosin's use for other PTSD hyperarousal symptoms (e.g., flashbacks, intrusive thoughts) and other substance use disorders, including nicotine and alcohol use disorders.

## Introduction

Doxazosin is an alpha-1 adrenergic antagonist currently approved by the United States Food and Drug Administration (FDA) for the treatment of benign prostatic

**Data availability statement:** All data can be found in the manuscript and supporting information files.

**Funding:** The authors received no specific funding for this work.

**Competing interests:** The authors have declared that no competing interests exist.

hyperplasia (BPH) and hypertension [1]. Over the past decade, doxazosin has been increasingly recognized as the me-too drug to prazosin, an alpha-1 adrenergic antagonist commonly used off-label for the treatment of post-traumatic stress disorder (PTSD) hyperarousal symptoms (e.g., flashbacks and nightmares), and less commonly substance use disorders. That being said, research on doxazosin for these conditions has been limited.

Stress in mental health disorders is associated with increased norepinephrine turnover in the brain and dysfunction of the prefrontal cortex, leading to impaired working memory, impulsiveness, and inappropriate behavioral regulation (Fig 1) [2]. While there is also peripheral activation of alpha-1 adrenoceptors by release of norepinephrine leading to the classic "fight or flight" response to stress, alteration in the hypothalamic-pituitary-adrenal (HPA) axis and sensitivity of central nervous system alpha-1 adrenoceptors is primarily thought to be implicated in several mental health conditions including depression, anxiety, and PTSD (Fig 1) [2–4]. Alpha-1 adrenergic antagonism is theorized to work physiologically in trauma and stress responses by attenuating the increased noradrenergic response on alpha-1 adrenoceptors at the prefrontal cortex [2–4].

Prazosin has been broadly examined for its psychiatric use, especially for PTSD, but doxazosin's longer half-life of 22 hours allows for once daily dosing to target full day symptoms, while prazosin's actions (2–3 hour half-life) may not last throughout the night or day [1,5–7]. In addition to immediate release (IR) starting at 1 mg, doxazosin also has a gastrointestinal therapeutic system (GITS) or extended release (XL) formulation starting at 4 mg with a reported lower side effect profile due to a lower serum peak-to-trough ratio [1]. Prazosin has historically been the focus of clinical trials due to its solubility properties and theorized ability to cross the blood brain barrier more easily than other alpha-1 antagonists; however, reported uses of doxazosin for various psychiatric conditions do suggest its ability to penetrate the blood-brain barrier, possibly due to psychological and physiological stress from psychiatric conditions leading to inflammation and increased permeability [8].

While prazosin has been the primary alpha-1 antagonist studied for psychiatric conditions in the literature, doxazosin has had additional interest recently due to its longer half-life, thought to alter sympathetic nervous system responses during the day without the need for multiple doses. Given the connection between the HPA axis and mental health, doxazosin has the potential to affect many mental health disorders. We present a scoping review to identify research about how doxazosin is being used in the treatment of any mental health disorder included in the Diagnostic and Statistical Manual of Mental Disorders-5th Edition (DSM-5), evidence of effectiveness for these disorders, and promising areas for further research.

## Methods

### Search criteria

The authors followed Preferred Reporting Items for Systematic Reviews and Meta-Analyses extension for Scoping Reviews (PRISMA-ScR) guidelines to complete this scoping review (S1 PRISMA Checklist) [9]. The research question did not utilize PICO

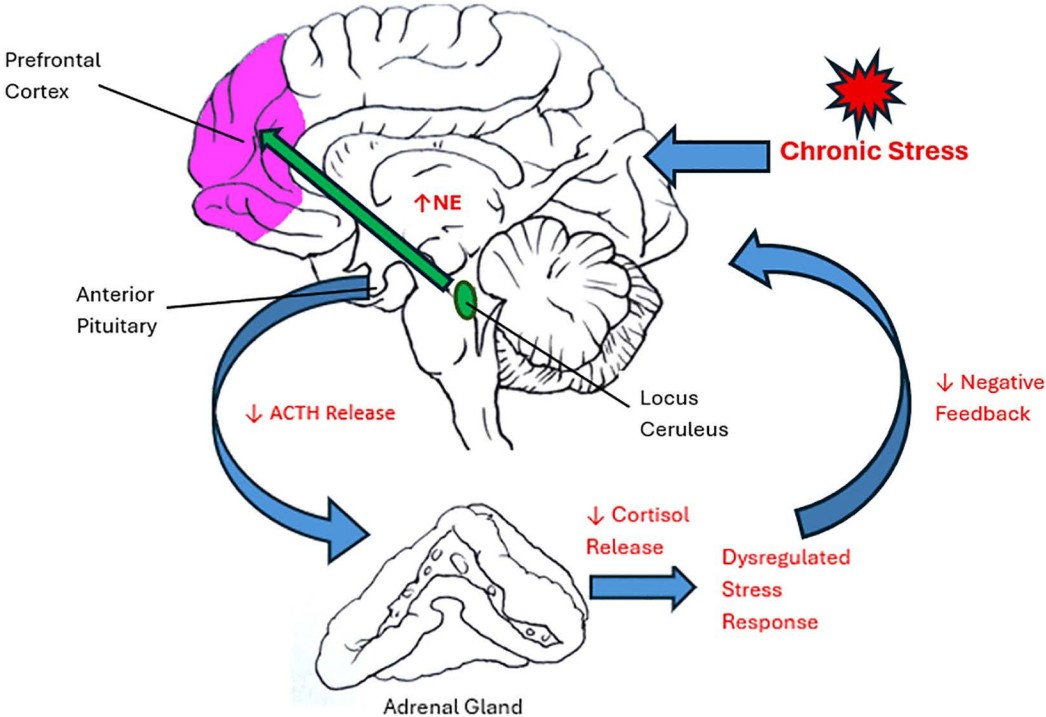

**Fig 1. Diagram of the effects of chronic stress on the prefrontal cortex and the HPA axis.**

guidelines to avoid limiting the clinical question, and in order to capture the breadth of the literature and identify key concepts. The search was conducted on November 8, 2024, through PubMed, Embase, PsycINFO, and Cochrane Central Register of Controlled Trials (CENTRAL) databases to identify articles in major health/mental health databases and to identify ongoing clinical trials. The primary keyword search used in retrieval of articles was: Doxazosin AND (Substance Use Disorder OR Substance Related Disorder OR Drug Use Disorder OR Drug Use OR Substance Abuse OR Drug Abuse OR Substance Dependence OR Drug Dependence OR Chemical Dependence OR Addiction OR relapse OR alcohol OR cocaine OR opioids OR nicotine OR hallucinogen OR sedative OR steroid OR cannabis OR cannabinoid OR posttraumatic OR post-traumatic OR PTSD OR depression OR anxiety OR mood OR mental health OR mental disorder OR psych). Chosen keywords were based on an initial literature review of mental health conditions treated by doxazosin and keywords used in the search for relevant article references.

## Eligibility criteria and data collection

During the first literature review, the lead author (CR) performed automated searches of the keywords in the included databases and removed duplicate articles, managed and automated in EndNoteTM21 software. CR manually reviewed the removed duplicate articles and screened the titles and abstracts for relevance to doxazosin used for mental health disorders (included in DSM-5). During the second literature review, all study members (CR, DA, AK, AB, PS) reviewed full texts for inclusion and exclusion criteria.

Inclusion criteria included doxazosin used for the treatment of any DSM-5 mental health disorder or related symptom(s), including depression, anxiety, and nightmares. Only articles in the English language were included. There was no cutoff for article dates published due to doxazosin for mental health disorders being a relatively new practice, and to be more inclusive. Exclusion criteria included articles not in the English language or not related to doxazosin treatment of a mental health disorder. Other excluded articles included animal trials, systematic reviews and/or meta-analyses, abstracts only, or full results

not being available, or commentaries. If there were any discrepancies about inclusion or exclusion criteria, the articles were reviewed with the lead author (CR) for consensus. The authors reviewed the articles and pulled data for review, including study design, participant demographics, primary outcome measures, doxazosin dose and adverse effects, and major findings. Final screening of the articles included in the final review process was completed by the lead author (CR).

## Results

Using PRISMA-ScR guidelines, 23 articles met full inclusion criteria and were included in the review (Fig 2).

### PTSD and/or nightmares

Nine articles evaluated doxazosin for PTSD and/or nightmares (Table 1) [10–18]. Doxazosin doses ranged from 1-16 mg per day. Only one RCT was identified, which found a significant reduction in some PTSD symptoms with doxazosin treatment compared to placebo [10]. Two open-label trials found a reduction in PTSD symptoms and sleep disturbance with doxazosin [11,12]. A retrospective chart review found a significant reduction in trauma-associated nightmares with

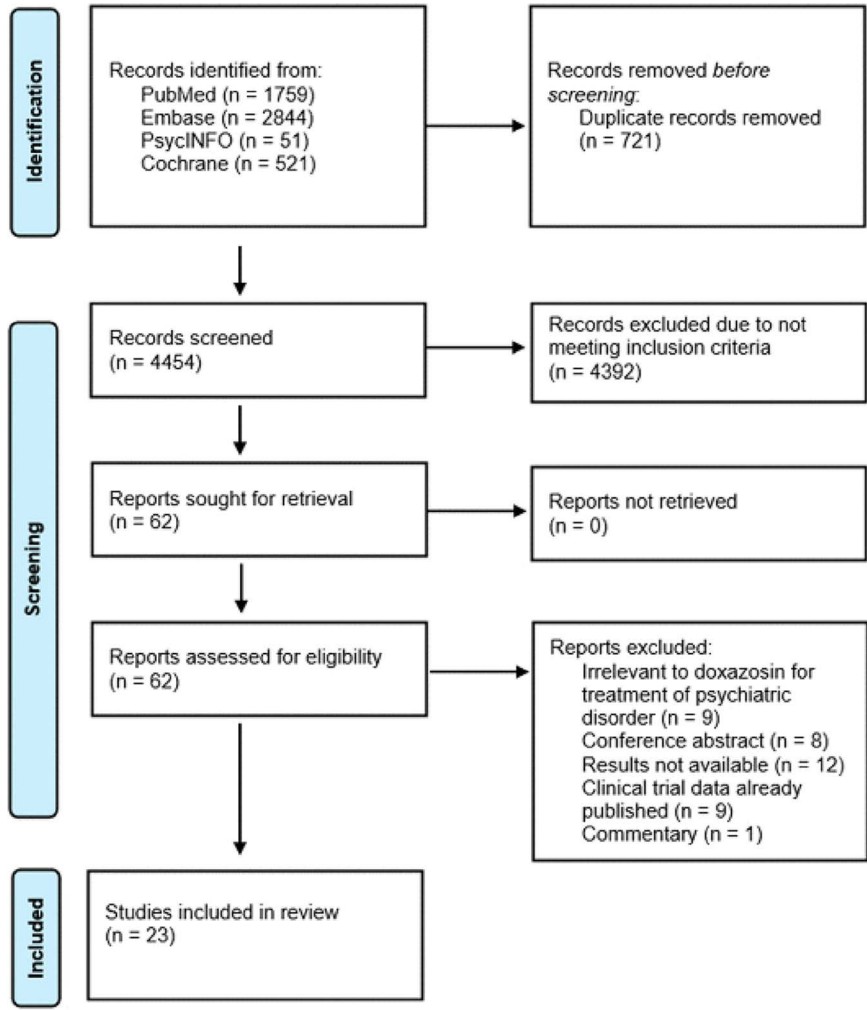

**Fig 2. PRISMA 2020 flow diagram.**

**Table 1. Summary of included studies of doxazosin used for PTSD and/or nightmares.**

| Reference | Study design | Patient demographics | Primary outcome measures | Doxazosin dose | Adverse effects | Major findings | Comments |
|---|---|---|---|---|---|---|---|
| *Randomized-controlled studies* | | | | | | | |
| Rodgman et al., 2016 [10] | Double-blind, placebo-controlled, within-subjects trial | 8 male veterans with PTSD based on military service, mean age 34.8 years | CAPS, PCL-M | 16 mg/day | Rhinitis | CAPS: No significant effects of treatment ($F_{1,13}=0.00$, P=.992) or time ($F_{1,13}=1.70$, P=.215) and no significant treatment x time interaction ($F_{1,13}=0.00$, P=.978). PCL-M: No significant effects of treatment ($F_{1,13}=6.33$, P=.026) and a significant treatment x time interaction ($F_{1,13}=15.93$, P=.002). | PTSD was diagnosed with DSM-IV criteria. Doxazosin XL formulation was used. No significant treatment x time interactions for PSQI, BAI, or BDI scores. 16-day medication treatment period. |
| *Open-label studies* | | | | | | | |
| De Jong et al., 2010 [11] | Open-label study | 12 participants with PTSD and severe disruptive sleep, mean age 37.9±9 years | CAPS, Montgomery-Asberg Depression Scale, CGI, pulse/tension control | 4 or 8 mg/day | Drowsiness | Decreases in CAP scores were significant at weeks 8 and 12 (P=0.006 and 0.005, respectively). Decreases in Montgomery-Asberg Depression Scale and CGI scores were significant at week 12 (P=0.046 and 0.023, respectively). No significant change in heart rate/blood pressure. | 4 male and 8 female participants. Three participants had adverse effects at 8 mg and continued the study at 4 mg. |
| Richards et al., 2018 [12] | Open-label study | 8 participants with full or partial PTSD ≥ 3 months, CAPS-IV score ≥ 30, distressing dreams score ≥ 5, age 40.1 ± 14.8 years | CAPS, PCL, PSQI, sleep diary | 4 or 8 mg/day | Drowsiness, low energy, weakness, nausea, nasal congestion, headache, subjective palpitations, increased heart rate | CAPS total, distressing dreams, and sleep disturbance scores dropped from 57.3 (SD=8.1) to 31.5 (SD=14.5) (z=2.52, p=.012); 5.5 (SD=.78) to 2.4 (SD=2.6) (z=2.49, p=.013); and 6.6 (SD5.91) to 4.3 (SD=3.2) (z=1.60, p=.110), respectively. Significant improvement on the PCL (average drop of 18.7 points; SM=10.88, p=.002) and PSQI (average drop of 4.1 points; SM=2.00, p=.030). The sleep diary indicated a significant reduction in nightmares (z=22.8, p=.006) and improvement in sleep quality (SM=3.1, p=.014). | Doxazosin XL formulation was used. BDI scores had a significant decrease in points. |

*(Continued)*

**Table 1.** (Continued)

| Reference | Study design | Patient demographics | Primary outcome measures | Doxazosin dose | Adverse effects | Major findings | Comments |
|---|---|---|---|---|---|---|---|
| *Retrospective studies* | | | | | | | |
| Roepke et al., 2016 [13] | Retrospective chart review | 51 German participants in an inpatient psychiatry unit diagnosed with PTSD and/or BPD and treated with doxazosin for trauma-associated nightmares. The mean age 35.7 years, 92.3% women. | CAPS B2, PSQI-A, sleep logs | The mean dose at week 12 was 6.08 mg (SD 3.30) | Unspecific "cardiac and circulatory disturbance" and more intense dreams | CAPS B2: Significant effect of time ($F_{(2,70)}$ = 11.55, $p < 0.001$) with a medium pre-post effect size ($d = 0.78$). PSQI-A: Significant effect of time ($F_{(2,56)}$ = 8.63, $p = 0.001$)). Sleep logs: Patients treated with doxazosin significantly improved in the items "recuperation of night sleep" (from mean 3.84 (SD = 0.69) to mean 3.46 (SD = 0.89), $t(31)$ = 3.42, $p = 0.002$) and "time awake at night" (from 74.67 min (SD = 70.60) to 66.97 min (SD = 62.63), $t(27)$ = 2.01, $p = 0.045$). | Prazosin was not available in Germany the month before the chart review. DSM-IV criteria were used for PTSD and BPD diagnoses. 23% of participants received trauma-focused CBT. |
| *Case series/reports* | | | | | | | |
| Pallesen et al., 2020 [14] | Case report | 56-year-old woman with ICD-10 criteria for PTSD due to a medical incident | Diary-based self-report | 4-8 mg/day | Dizziness in hot weather and rising quickly from lying positions, one occurrence of fainting while taking 8 mg | Nightmares were absent in 21.6% of days with 4 mg of doxazosin and in 55.2% with 8 mg of doxazosin. Adjusted model was statistically significant ($\chi2 = 45.7$, $df = 3$, $p < 0.001$) | Nightmares occurred 11 months before starting doxazosin. Diary about nightmares with doxazosin doses kept for 280 days. Blood pressure was not abnormal during any clinical evaluations. |
| Calegaro et al., 2019 [15] | Case report | 3 middle-aged Brazilian males with MDD who were diagnosed with PTSD after nightclub fire in 2013 | Clinical interview | 2-4 mg/day | Not reported | Case 1: Doxazosin resolved nightmares but did not improve insomnia. Case 2: Doxazosin resolved nightmares and insomnia. Case 3: Doxazosin reduced nightmares to less than once a week | Case 1: Concurrently prescribed venlafaxine 225 mg, quetiapine 125 mg, and chlorpromazine 200 mg. Case 2: Concurrently prescribed sertraline 200 mg, Chlorpromazine 300 mg, and clonazepam 2 mg. Case 3: Concurrently prescribed sertraline 200 mg, lithium carbonate 1,500 mg, and clonazepam 1 mg |

*(Continued)*

**Table 1.** (Continued)

| Reference | Study design | Patient demographics | Primary outcome measures | Doxazosin dose | Adverse effects | Major findings | Comments |
|---|---|---|---|---|---|---|---|
| Sethi and Vasudeva, 2012 [16] | Case report | 59-year-old veteran with DSM-IV-TR criteria for PTSD | PCL-M | 4 mg/day | Orthostatic hypotension in the initial days after starting therapy | PCL-M scores (weeks 1, 4, 8) decreased from 5 to 2 for disturbing dreams, 4–2 for disturbing memories, thoughts, or images | Concurrently treated with hydrochlorothiazide for hypertension and sertraline for MDD. Trials for insomnia included trazodone, mirtazapine, and quetiapine. |
| Khan et al., 2024 [17] | Case report | 3 adults (63 years old, 72 years old, 73 years old) with PTSD-related nightmares | Clinical interview | 1 mg/day, 4 mg/day, and 8 mg/day | "Blackout" and a fall, unclear if related to doxazosin | Case 1: Participant reported complete resolution of nightmares and flashbacks for more than 2 years on doxazosin Case 2: Participant reported complete resolution of nightmares for more than a year on doxazosin Case 3: Participant reported complete resolution of nightmares for a month on doxazosin | Case 1: Concurrently prescribed sertraline 250 mg, melatonin 5 mg, quetiapine 200 mg, aripiprazole 10 mg Case 2: Concurrently prescribed quetiapine 75 mg, diazepam 40 mg, doxepin 100 mg Case 3: No listed medications concurrently prescribed. Doxazosin IR formulation used. |
| Hori, 2021 [18] | Case report | 71-year-old Japanese female with MDD in remission and new (non-PTSD) nightmares associated with the COVID-19 pandemic | Clinical interview | 1 mg/day | None reported | Nightmares remitted, and blood pressure decreased from 145/95 mmHg to 125/80 mmHg in 2 weeks. Nightmares quickly recurred when doxazosin was discontinued for 4 days, then completely resolved within 2 weeks after restarting. | MDD in remission on escitalopram 10 mg daily and eszopiclone 1 mg HS for 2 years before new nightmares. Initial unsuccessful trials for nightmares included trazodone 50 mg HS, eszopiclone 2 mg HS, and risperidone 1 mg HS. |

PTSD: Post-traumatic stress disorder; ICD-10: International Classification of Diseases, 10th revision; MDD: Major Depressive Disorder; HS: Nightly; BPD: Borderline personality disorder; CAPS: Clinician-Administered PTSD Scale; CGI: Clinical Global Impression; PSQI-A: Pittsburgh Sleep Quality Index Addendum for PTSD; CBT: Cognitive behavioral therapy; PCL-M: Posttraumatic Stress Disorder Checklist Military version; BAI: Beck Anxiety Index; BDI: Beck Depression Index; DSM: Diagnostic and Statistical Manual of Mental Disorders.

doxazosin in patients with a primary diagnosis of either PTSD or borderline personality disorder [13]. Four case reports illustrated either complete remission (no further reported nightmares during the study period) or significant reduction of nightmares in patients with a PTSD diagnosis [14–17]. In one case report, non-trauma-related nightmares resolved with doxazosin in a patient with a primary diagnosis of major depressive disorder [18].

## Substance use disorders

Twelve articles evaluated doxazosin for the treatment of substance use disorders (Table 2) [19–30]. Doxazosin doses ranged from 4-16 mg per day. All articles were RCTs (two clinical trials without associated publications). Seven of the articles studied doxazosin for cocaine use disorder (CUD), with all except for two studies (one being clinical trial data without an associated publication) showing a reduction in cocaine use with doxazosin [19–25]. One article evaluated doxazosin

**Table 2. Summary of included studies of doxazosin used for substance use disorders.**

| Reference | Study design | Patient demographics | Primary outcome measures | Doxazosin dose | Adverse effects | Major findings | Comments |
|---|---|---|---|---|---|---|---|
| *Cocaine use disorder* *Randomized-controlled studies* | | | | | | | |
| Shorter et al, 2020 [31] | Double-blind, randomized, placebo-controlled pilot trial | 76 participants with DSM-IV criteria for cocaine dependence, divided by $\alpha_1$ adrenoceptor subtype D (ADRA1D) gene polymorphism (AA vs AT/TT) | Urine toxicology | 8 mg/day | One patient developed a nonspecific rash/hives, determined to be due to metronidazole, and was discontinued from the study | Doxazosin treatment reduced cocaine use in the AT/TT group compared to placebo, and demonstrated a medication effect (F = 13.0; df = 1,769; p = 3.3 x 10−4, η2 = 0.017) and time by medication effect (F = 11.2; df = 1,769; p = 8.6 x 10−4, η2 = 0.015). Doxazosin showed no treatment difference from placebo in the AA group. | CUD was used in the article for scientific consistency based on DSM-5 diagnostic criteria. 73% were male, 72% were African American. No differences in baseline cocaine cravings between groups. There was a higher percentage of African Americans in the AA group vs AT/TT group (84% vs. 50%). Number of missing at random urines was similar among the groups. |
| Zhang et al., 2018 [19] | Double-blind, randomized, placebo-controlled trial | 76 participants with DSM-IV criteria for cocaine dependence, divided by dopamine β-hydroxylase (DβH) polymorphism (CC vs CT/TT) | Urine toxicology | 8 mg/day | One patient developed a nonspecific rash/hives, determined to be due to metronidazole, and was discontinued from the study | Doxazosin treatment reduced cocaine use with greater efficiency in the CT/TT group (medication effect, F = 7.38, df = 1, 518; p = 0.007, η2 = 0.014; time by medication, F = 5.92, df = 1, 518; p = 0.015, η2 = 0.011) than the CC group (time by medication, p > 0.05). | CUD used in the article for scientific consistency based on DSM-5 diagnostic criteria. No differences in baseline cocaine cravings between groups. Higher percentage of African Americans in the CC group vs. CT/TT group (88% vs. 53%). Number of missing at random urines was similar among the groups. |
| Nielsen et al., 2017 [20] | Double-blind, randomized, placebo-controlled pilot trial (abstract) | 76 participants with DSM-IV criteria for cocaine dependence, assessed for adrenoreceptor genetic variants (ADRA1A rs1048101, ADRA1B rs3729604, and ADRA1D rs2236554) | Urine toxicology | 8 mg/day | One patient developed nonspecific rash/hives, determined to be due to metronidazole, discontinued from the study. | Doxazosin group had 12% decrease in cocaine positive urines, while placebo group had 11% increase (p < 0.0001). The ADRA1B rs3729604 GG genotype subjects (p < 0.0001) and the ADRA1D rs1048101 T-allele carriers (p = 0.046) responded to doxazosin treatment, while those of the corresponding genotype groups did not. | CUD used in the article for scientific consistency based on DSM-5 diagnostic criteria. Participants used cocaine for mean of 18 years, mean age 48 years, 73% were male, 72% African American. |

*(Continued)*

**Table 2.** (Continued)

| Reference | Study design | Patient demographics | Primary outcome measures | Doxazosin dose | Adverse effects | Major findings | Comments |
|---|---|---|---|---|---|---|---|
| Shorter et al., 2013 [21] | Double-blind, randomized, placebo-controlled pilot trial | 35 participants with DSM-IV-TR diagnosis of cocaine dependence, self-reported cocaine use ≥ once weekly for ≥ one month, cocaine-positive urine, ≥ 3 on SDS | Urine toxicology | 8 mg/day | Most common were headache, dry mouth, tiredness, and nausea/vomiting | Total number of cocaine-negative urines was significantly increased in the DOX-fast group (35%) vs. the DOX-slow group (10%) and placebo group (14%) (chi-square = 36.3, df = 2, p < 0.0001). Percentage of participants achieving ≥ 2 consecutive weeks of abstinence 0% for the DOX-slow group, 44% for the DOX-fast group, and 7% for placebo (chi-square = 7.35, df = 2, p = 0.025). | Participants given doxazosin were titrated to target dose by week 4 (DOX-fast) or week 8 (DOX-slow). Participants were predominantly African American (57%). |
| Mancino et al., 2015 [22] | Double-blind, randomized, placebo-controlled trial | 14 cocaine (mean age 44.3 ± 7.6 years) and 8 methamphetamine (mean age 34.5 ± 6.2 years) dependent participants | Urine toxicology | 8 mg/day | Palpitations, shortness of breath, heartburn, fatigue | Cocaine and methamphetamine positive urines significantly increased over time with doxazosin treatment relative to placebo (OR=1.05, z=2.20, p < 0.03) | Cocaine dependent participants were more likely to be African American (71% COC vs 12.5% METH, p = 0.02). Doxazosin XL formulation used. |
| Newton et al., 2012 [23] | Single-site, randomized, placebo-controlled, within-subjects study | 13 participants with DSM-IV diagnosis of cocaine dependence, non-treatment seeking, mean age 44.31 ± 4.63 years | BP, HR, subjective effects of cocaine | 4 mg/day | None reported | Doxazosin had trend-level effects on systolic BP following cocaine dosing (p=.09) but did not significantly affect diastolic BP or HR. Doxazosin reduced the effects of 20 mg cocaine on ratings of "stimulated" (t=2.20, p=.035), "like" (t=2.16, p=.037), and "likely to use cocaine if had access" (t=2.29, p=.028). Doxazosin produced trend-level reductions in the effects of 40 mg cocaine on ratings of "stimulated" (t=1.78, p=.083), "desire cocaine" (t=1.96, p=.059), and "likely to use cocaine if had access" (t=1.78, p=.083). | Doxazosin IR formulation used. 92% male, 69% African American. |
| Shorter 2020 (S1 Data) [24] | Double-blind, randomized, placebo-controlled trial (clinical trial without associated publication) | 43 participants with DSM-IV diagnosis of cocaine dependence, self-reported use of cocaine within last 90 days or ≥ 1 cocaine positive urine during screening, mean age 54.2 years | Urine toxicology | 8 mg/day | Most common was nausea | Average percentage of cocaine positive urines over 12 weeks was 67.6% for doxazosin group and 69% with placebo group | 95.3% male, 83.7% African American. |

*(Continued)*

**Table 2.** (Continued)

| Reference | Study design | Patient demographics | Primary outcome measures | Doxazosin dose | Adverse effects | Major findings | Comments |
|---|---|---|---|---|---|---|---|
| *Nicotine use disorder* *Randomized-controlled studies* | | | | | | | |
| Roberts et al., 2018 [25] | Double-blind, randomized, placebo-controlled trial | 35 non-treatment-seeking adult smokers who smoked ≥10 cigarettes/day for past year | CPT, smoking lapse test, MNWS | 4 or 8 mg/day | Most common were fatigue, headache, dizziness, and drowsiness | Only the 8 mg/day group decreased commission errors during the deprivation session, $t(12) = 2.00$, $p = 0.04$. No sig. difference in MNWS scores between sessions with 4 mg/day doxazosin, $t(10) = 0.92$, $p = 0.19$. Those with 8 mg/day had sig. increase in MNWS scores during deprivation session $(12) = 2.01$, $p = 0.03$. | Participants receiving active doxazosin reported fewer withdrawal symptoms; those with fewer withdrawal symptoms could resist smoking for longer and smoked less when given free access to cigarettes. |
| Verplaetse et al, 2017 [26] | Double-blind, randomized, placebo-controlled pilot trial | 35 adults who smoked 10 cigarettes/day for past year, urine cotinine levels of ≥150, normotensive | Questionnaire of Smoking Urges-Brief, latency to smoke, ad-libitum smoking | 4 or 8 mg/day | Most common were fatigue, headache, dizziness, and drowsiness | Doxazosin 8 mg/day decreased tobacco craving following stress imagery relative to placebo ($F(2,26) = 4.56$, $p = 0.02$; Cohen's $d = 0.84$). Doxazosin increased the latency to start smoking in the stress versus neutral imagery condition only ($F(1,32) = 4.57$, $p = 0.04$; Cohen's $d = 0.76$). Doxazosin decreased the number of cigarettes smoked during the 60-minute ad-libitum smoking period relative to placebo ($F(1,30) = 4.01$, $p = 0.05$; Cohen's $d = 0.73$) | Additional measures included cortisol, ACTH, norepinephrine, epinephrine, and physiological measures. Doxazosin 8 mg/day increased cortisol levels following stress imagery and decreased cortisol levels following neutral imagery, and attenuated increases in epinephrine levels in the stress versus neutral imagery condition. Systolic blood pressure increased from pre- to post-imagery in the placebo vs. doxazosin group, and doxazosin decreased this effect post-imagery. |
| *Alcohol use disorder* *Randomized-controlled studies* | | | | | | | |
| Haass-Koffler et al., 2017 [27] | Double-blind, randomized, placebo-controlled trial | 41 participants with DSM-IV diagnosed alcohol dependence seeking treatment, heavy drinking during 90-day period before screening | Standing BP | 16 mg/day (or maximum-tolerable dose) | Most common in doxazosin group included dizziness, depression or other mood disturbance, trouble urinating, headache | Significant medication effect for higher diastolic BP in the hypothesized direction (i.e., reduction of drinking) for both DPW [$t32 = -2.299$, *$p = 0.028$] and HDD [$t32 = -3.216$, **$p = 0.003$]. No effects on standing systolic BP × medication interaction ($p$'s > 0.05). | Individuals with higher diastolic BP had less of a reduction in craving. Standing diastolic BP was not related to FHDA. |

*(Continued)*

**Table 2.** (Continued)

| Reference | Study design | Patient demographics | Primary outcome measures | Doxazosin dose | Adverse effects | Major findings | Comments |
|---|---|---|---|---|---|---|---|
| Kenna et al., 2016 [28] | Double-blind, randomized, placebo-controlled trial (proof of concept) | 41 participants with DSM-IV diagnosed alcohol dependence seeking outpatient treatment, mean age 42.1 years | DPW, HDD | 16 mg/day (or maximum tolerable dose) | Most common in doxazosin group included dizziness, depression or other mood disturbance, trouble urinating, headache | No significant differences between groups in DPW [F1,36 = 0.43, p > .05] and HDD [F1,35 = 1.03, p > .05], although there was a small reduction in DPW and HDD in the doxazosin group compared to placebo (effect sizes d = .23 and .35, respectively) | There was a significant main effect for medication on the ODS subscale. There was a significant medication effect for high FHDA in reducing drinking for DPW and HDD. |
| Curtin, 2021 (S2 Data) [29] | Double-blind, randomized, placebo-controlled trial (clinical trial without associated publication) | 61 participants with alcohol use disorder in early abstinence, mean age 42.5 ± 11.6 years | Startle potentiation, TLFB | 8 mg/day | Most common were dizziness, fatigue, rhinitis, headache, dry mouth | Unpredictable startle potentiation mean for doxazosin was 22.25 (SD 20.18) vs placebo 20.76 (SD 23.21). Predictable startle potentiation mean for doxazosin was 26.22 (SD 30.90) vs placebo 22.21 (SD 35.93). Heavy drinking reported in 58.6% of doxazosin participants vs 65.6% of placebo participants. | 75.4% male, 88.5% White race. Only 26 participants completed NPU tasks. |

PTSD: Post-traumatic stress disorder; ICD-10: International Classification of Diseases, 10th revision; CUD: Cocaine use disorder; AUD: Alcohol use disorder; MNWS: Minnesota Nicotine Withdrawal Scale; ACTH: Adrenocorticotropic hormone; DPW: Drinks per week; HDD: Heavy drinking days; ODS: Obsessive drinking scale; FHDA: Family history density of alcoholism; SDS: Severity of Dependence Scale; DSM: Diagnostic and Statistical Manual of Mental Disorders; BP: Blood pressure; HR: Heart rate; NPU: No-shock, predictable shock, unpredictable-shock; TLFB: Time-line follow-back.

for both cocaine and methamphetamine, which showed increased positive urine toxicology screens with doxazosin use [23]. Two articles found that doxazosin reduces cravings for cigarette smoking [26,27]. The remaining three articles evaluated doxazosin for the treatment of alcohol use disorder (AUD) [28–30].

## Dual diagnoses

Two articles evaluated doxazosin for the treatment of dual diagnosis conditions (Table 3) [30,32]. Doxazosin doses ranged from 1-16 mg per day. An RCT evaluated doxazosin for veterans with co-occurring PTSD and AUD and found no improvement in PTSD scores with doxazosin but an increased likelihood to abstain from alcohol compared to placebo [30]. A case report for a female with PTSD and AUD described continued successful treatment of PTSD symptoms when switched from prazosin to doxazosin [32].

## Discussion

Currently, there is evidence for using doxazosin to treat substance use disorders (cocaine, methamphetamine, nicotine, alcohol), sleep quality in PTSD, and nightmares, although recommendations are limited by a lack of high-quality RCTs. There is greater strength of evidence for doxazosin in the treatment of substance use disorders, specifically CUD, but it appears to be associated with specific genotypes. There is surprisingly a lack of RCTs for PTSD and nightmares, considering a significant amount of attention by the VA on treating military trauma-related nightmares with prazosin. Although most of the research on alpha-1 antagonists for PTSD and substance use disorders has been done with prazosin, there

**Table 3. Summary of included studies of doxazosin used for dual diagnosis.**

| Reference | Study design | Patient demographics | Primary outcome measures | Doxazosin dose | Adverse effects | Major findings | Comments |
|---|---|---|---|---|---|---|---|
| Back et al., 2023 [30] | Double-blind, randomized, placebo-controlled trial | 92 veterans with co-occurring PTSD and AUD, 84.4% male, mean age $45 \pm 13$ years | CAPS-5, PCL-5, TLFB | 1 – 16 mg/day | Common adverse events included dizziness, gastrointestinal symptoms, joint/muscle pain, cold or sinus congestion, sleep problems, and vivid dreams/nightmares | No significant difference in CAP-5 and PCL-5 scores between doxazosin and placebo. Participants in the doxazosin group were more likely to abstain from alcohol than placebo group (22% vs 7%, $X^2_1 = 5.7$, p = .017); however, they consumed a greater number of drinks on the days they consumed alcohol. | Study population may not generalize to other populations, primary outcome measures were self-reports. Doxazosin IR formulation used. |
| Hessel et al, 2022 [32] | Case report | 43-year-old female with diagnosis of AUD and PTSD treated with prazosin 2 mg nightly | Clinical interview | 4 mg/day | Not reported | Prior reported PTSD symptoms (hypervigilance, nightmares, anxiety, intrusive memories, insomnia) resolved with prazosin 2 mg did not recur when switched to doxazosin 4 mg | Lack of details about PTSD and AUD symptoms while taking prazosin |

PTSD: Post-traumatic stress disorder; CAPS: Clinician-Administered PTSD Scale; AUD: Alcohol use disorder; TLFB: Time-line follow-back; PCL-5: PTSD Checklist for DSM-5.

is increasing interest in doxazosin due to its significantly longer half-life, which theoretically would increase tolerability. It is challenging to comment on the tolerability differences of prazosin and doxazosin without head-to-head studies. Common side effects of alpha-1 antagonists include orthostatic hypotension, dizziness, falls, and syncope, so theoretically, dosing a medication once daily would reduce the risk of these adverse effects. However, a recent article evaluating rates of post-market side effects found that doxazosin had higher rates of dizziness and syncope compared to prazosin, tamsulosin, and terazosin [33]. Although there are no recent head-to-head trials, an older RCT comparing prazosin to doxazosin for hypertension did not note tolerability differences [34]. Doses used in other conditions, including hypertension and BPH, are higher than those used in studies for mental health conditions, so a conclusive statement about tolerability would be challenging without further research on this [1].

A limitation of this review is that articles were not assessed or screened for risk of bias, as the goal was to understand trends for further potential research, including low-quality research such as case reports. Of note, there was only one RCT published about doxazosin for PTSD, but the quality of the study was lowered by having a small sample size (8 participants) and poor generalizability (only male veterans). The rest of the studies for PTSD consist of low-quality studies, including open-label trials, a retrospective review, and case series/reports. There was a higher quality of evidence for substance use disorders with 12 RCTs, but the quality was lowered with poor generalizability in most studies (majority male participants; majority African American participants in all CUD studies). Finally, there was inherent selection bias as only four databases were searched for articles, and it is possible that other literature could have been missed.

## PTSD and/or nightmares

A recent systematic review and meta-regression analysis of prazosin found significant improvement in insomnia and nightmares but not overall PTSD symptoms [35]. The RCT by Rodgman et al. studying doxazosin for PTSD did show significant improvement in the PTSD Checklist for DSM-5 (PCL-M) but not in the Clinician-Administered PTSD Scale (CAPS) or depression scales [10]. The authors did note that CAPS is clinician-administered and more intensive, which could trigger more memories and symptoms compared to PCL, which is self-administered [10]. In addition, the study was very short

(16-day treatment period) compared to prior prazosin studies (typically 8–15 weeks), the sample size was small (n = 8), and participants were limited to male veterans compared to other PTSD open-label studies and a retrospective review with prazosin in which most of the participants were female [10,36,37]. The two open-label studies studying doxazosin for PTSD found a significant reduction in CAPS, and one saw reductions in PCL and Pittsburgh Sleep Quality Index (PSQI); however, both studies had a small number of participants complete the full trial, with many dropping out due to side effects, especially drowsiness [11,12]. The retrospective chart review also found a reduction in CAPS (B2) and PSQI-A scores in patients with either PTSD or borderline personality disorder [13]. Complicating the picture further, an RCT of doxazosin for comorbid PTSD and AUD diagnoses found a reduction in both CAPS and PCL scores, but not significantly compared to placebo [30]. Notably, though, this study only looked at male veterans as well, indicating participant demographics may influence response and limited generalizability. The five case reports on doxazosin for PTSD and/or nightmares only reported improvement of nightmares (one report with a non-PTSD diagnosis), indicated by clinical interview, PCL-5, and diary-based self-report. The current trials and case reports on doxazosin for PTSD and/or nightmares suggested that doxazosin reduces the intensity and frequency of nightmares and can improve sleep quality in patients with PTSD, similar to findings for prazosin; however, the current evidence is limited to small participant trials and case reports [10–18].

Prazosin has been suggested to reduce daytime PTSD symptoms in addition to nightmares, including re-experiencing (flashbacks) and hypervigilance to trauma reminders when dosed multiple times a day [38–40]. Two of the three clinical trials for treating PTSD symptoms with doxazosin did not specifically document a reduction in daytime symptoms. While the two open-label studies in this review reported a statistically significant reduction in CAPS with doxazosin, the only sub-score changes noted to be significant were related to nightmare reduction [11,12]. The retrospective review combined PTSD and borderline personality disorder diagnoses and only obtained the CAPS B2 score, which just evaluates for nightmare intensity and frequency [13]. The one RCT reported on daytime symptoms for CAPS, including reexperiencing, avoidance, and hyperarousal, which did improve with doxazosin but were not statistically significant [10]. One case report illustrated the reduction of daytime symptoms (disturbing memories, thoughts, or images of a stressful military experience) from 4 (quite a bit) to 2 (a little bit) at four and eight weeks with 4 mg/day of doxazosin [16]. Another case report also described complete resolution of daytime flashbacks with doxazosin 4 mg/day [17]. More studies should be done to evaluate for changes in PTSD daytime symptoms with a single dose of doxazosin.

In addition to IR, doxazosin is available as XL, also known as GITS. An RCT by Andersen et al. in 2000 evaluated the efficacy and tolerability of doxazosin IR and XL for BPH and found that both formulations were effective with similar incidence and types of adverse effects [41]. Doxazosin IR in this study had a slightly increased number of adverse events compared to XL, which likely drove future RCTs for other indications to consider XL as the formulation of choice [41]. The limitation of a study using doxazosin for BPH is that typically the dose must be closer to 8 mg to improve maximum urinary flow rates, while significantly lower doses have been effective for PTSD and nightmares. According to the FDA, there is theoretically less variation in plasma levels with doxazosin XL and thus fewer side effects, although the half-lives of doxazosin IR and XL are relatively the same, and both are well-tolerated with similar rates of adverse effects [1]. The RCT and two open-label studies in this review only utilized doxazosin XL, while the retrospective review and three of the case reports indicated improvement in nightmares with doxazosin IR. The participant dropout rate due to adverse effects was significant in the open-label trials, especially given the low number of participants, which may have been due to needing to start at a higher doxazosin dose (XL starts at 4 mg versus IR starts at 1 mg) and lack of tolerance to this dose [11,12]. Richards et al. noted in their open-label study with doxazosin XL that they only perceived the XL formulation to be advantageous due to its ease of initiation at a higher and more therapeutic dose, but also recognized that the limitation in dose ranges may have limited tolerability [12]. The retrospective review did not specify doxazosin IR or XL use, but IR was more likely primarily used due to reported doses under 4 mg [13]. Only one case report specified doxazosin IR use, and two others reported doses under 4 mg [15,17,18]. Reports describing only 4 mg or 8 mg could indicate the XL version

being utilized, as these are the only formulation doses available. Given concern for participant dropout rate due to adverse effects in the studies using doxazosin XL, there should be more studies utilizing doxazosin IR to assess if it is better tolerated due to having a lower needed dose and being able to titrate in smaller increments.

## Substance use disorders

Seven articles were identified that addressed doxazosin for CUD, all of which were RCTs. There has been increasing evidence in animal trials that peripheral noradrenergic mechanisms have a role in the mediation of stimulant effects, thus leading to studies of alpha-1 adrenergic antagonists to reduce the physiological effects of stimulants and cravings [42]. Animal trials suggest that prazosin may reduce cocaine cravings [43]. Doxazosin was likely chosen for the human trials in this review primarily for its once-daily dosing and longer half-life. A study by Newton et al. suggested that doxazosin IR 4 mg/day can reduce subjective cravings for cocaine, but it was a small trial [23]. Two RCTs suggested a correlation between adrenoceptor and dopamine-β hydroxylase polymorphisms and doxazosin's effect on cocaine use [19,20,31–]. Specifically, the RCT by Shorter et al. found that doxazosin reduced cocaine use in the AT/TT (adrenoceptor) and CT/TT (DβH) genotypes compared to the typical genotype and placebo, suggesting a genetic connection to doxazosin's effectiveness. A trial by Shorter et al. found that more quickly titrated doxazosin (over four weeks) was more effective at reducing cocaine use compared to a slower titration (over eight weeks), but the subject size was low, and the results have not been replicated [21]. An RCT by Mancino et al. found that doxazosin did not reduce use of both cocaine and methamphetamine, but the participant number was very low, and there was no differentiation by genotype or titration schedule [22]. One clinical trial data set did not indicate any reduction in cocaine use with doxazosin [24]. Of note, most participants in these CUD trials were African American males, which inherently limits generalizability.

Two RCTs with presumed same study participants addressed doxazosin for nicotine use disorder [25,26]. The study by Roberts et al. evaluated doxazosin's effect on cognitive functioning and found improvement on continuous performance tasks (CPT) during nicotine withdrawal, longer resistance to smoking, and less overall smoking with doxazosin compared to placebo [25]. Interestingly, only doxazosin 4 mg/day improved withdrawal symptoms, and participants who received 8 mg/day reported an increase in withdrawal symptoms, thought to be due to paradoxical sympathomimetic effects at higher doses or altered noradrenergic tone with increased susceptibility to stressors, namely nicotine withdrawal [25]. The study by Verplaetse et al. suggested that doxazosin can attenuate stress (delivered via personalized guided imagery) on tobacco cravings, reducing cigarette use, possibly by normalizing cortisol's response to stress in nicotine-deprived smokers [26]. These two articles did indicate that doxazosin may be used in smoking cessation, but were completed by the same institution on what appears to be the same study participants. More studies are needed to verify these results.

Three RCTs evaluated doxazosin for AUD (including one clinical trial without publication). Two of the RCTs utilized the same participant population and focused on patient characteristics affecting doxazosin's effects on alcohol use [27,28]. The parent study by Kenna et al. found preliminary data that doxazosin may be effective in reducing alcohol craving and consumption in those with high FHDA [28]. The following study by Haass-Koffler et al. suggested that standing blood pressure may be an independent biomarker of doxazosin's response in AUD, as higher diastolic blood pressure predicted a more robust response to doxazosin in reducing alcohol use [27]. A clinical trial without publication evaluated startle potentiation during a stress reactivity task (shock) for participants given doxazosin versus placebo, but did not appear to find any major differences and trended towards the doxazosin group having increased unpredictable startle potentiation [29]. In addition, reported heavy drinking scores between groups did not appear to be statistically significant, but other RCTs did indicate that participant features (genetics and blood pressure) may indicate doxazosin's effectiveness, which this trial did not stratify [29]. Current studies on doxazosin for AUD, especially for specific patient populations, are promising, but more RCTs should be completed to more confidently recommend doxazosin as part of AUD treatment.

Similar to doxazosin studies for PTSD and/or nightmares, it appears that most of the studies for substance use disorders likely utilized the doxazosin XL formulation. However, only two articles specified formulation (Mancino et al. used XL

and Newton et al. used IR [22,23]), so it is assumed that, because the other study methods only indicated starting doses of 4 mg, and maximum doses of 8 or 16 mg, the XL formulation was used. Adverse effects did not appear to lead to large dropout rates, possibly indicating higher tolerability in substance use disorder participants.

In comparison to studies about doxazosin for PTSD and nightmares, all the studies on substance use disorders had a significantly skewed population towards males, which is not surprising, as males are 2–3 times more likely to have a substance use disorder compared to females [44]. Another noted limitation that prevents generalizability is that most participants in the CUD studies were African American males, and most of the participants in the alcohol use disorder studies were white males. Although males are statistically more likely to have a substance use disorder, the use of substances in females has also been rising, leading to increasing rates of substance use disorders due to the telescoping effect, or the faster progression to substance dependence in females compared to males [45]. These points should be considered in future studies, as race and gender differences can change dosing and tolerability of medications.

### Dual diagnoses

Two articles addressed doxazosin for co-occurring PTSD and AUD. An RCT by Back et al. found a reduction in the number of drinking days with doxazosin IR, but with greater consumption on drinking days and no significant difference in PTSD scores [30]. This study only included veterans and may not be generalizable, and outcome measures were only self-reported. Results were consistent with similar studies with prazosin, which also showed a reduction in alcohol use but with no significant changes in PTSD scales [46]. Another study evaluating prazosin for PTSD and AUD in veterans found no benefit for alcohol reduction or PTSD symptoms [47]. A case report by Hessel et al. found that switching from prazosin to doxazosin in a patient with PTSD and AUD did not result in a recurrence of PTSD symptoms (most recently nightmares and insomnia) in a patient admitted for AUD-associated pancreatitis, but they did not discuss the effects of alpha-1 antagonists on alcohol use [32]. It was not documented whether the IR or XL formulation was used in the case report, but given that prazosin 2 mg was switched to doxazosin 4 mg, it can be assumed that the XL formulation was used, as there would be no indication for doubling the dose upon switching. Overall, there is not enough quality evidence to make any recommendations about doxazosin's use for dual diagnoses. AUD may alter doxazosin's effect on PTSD symptoms, especially with studies showing that doxazosin's effect on alcohol use may be moderated by genetics and other physiological factors, but this would need to be researched further.

### Conclusion

There are a limited number of publications with strong evidence for the use of doxazosin for mental health disorders, but the current literature is promising. Positive evidence is available for doxazosin's use in PTSD-related nightmares and cocaine use disorders, which appears to be the focus of current clinical trials. Most of the articles for PTSD are small open-label trials or case reports, which is a major limitation for recommendations. Doxazosin does seem to be well-tolerated in clinical trials, although this has not been explicitly compared to prazosin. More RCTs would be of benefit to further clarify how to use doxazosin for PTSD and substance use disorders. In addition, current literature on doxazosin for substance use disorders has found significance with genetic predisposing factors. As genetics and personalized medicine are becoming more prominent, further research on genetic and physiological characteristics to predict doxazosin's effectiveness would be of interest to define further.

### Supporting information

**S1 Prisma Checklist.  PRISMA Checklist Preferred Reporting Items for Systematic reviews and Meta-Analyses extension for Scoping Reviews (PRISMA-ScR) Checklist.**
(PDF)

**S1 Data. Data for active clinical trial by Shorter et al, 2020.**
(PDF)

**S2 Data. Data for active clinical trial by Curtin et al, 2021.**
(PDF)

## Author contributions

**Conceptualization:** Christie Richardon.

**Data curation:** Christie Richardon.

**Formal analysis:** Christie Richardon, Danah Atassi, Afshan Khan, Amanda Bernardini, Priyal Shah.

**Investigation:** Christie Richardon.

**Methodology:** Christie Richardon.

**Software:** Christie Richardon.

**Supervision:** Christie Richardon.

**Visualization:** Christie Richardon.

**Writing – original draft:** Christie Richardon, Danah Atassi, Afshan Khan, Amanda Bernardini, Priyal Shah.

**Writing – review & editing:** Christie Richardon, Danah Atassi, Afshan Khan, Amanda Bernardini, Priyal Shah.

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
