## [Decision Letter · Decision Letter 0]

19 Aug 2025

PMEN-D-25-00312

Doxazosin for the treatment of mental health disorders: A scoping review

PLOS Mental Health

Dear Dr. Richardon,

Thank you for submitting your manuscript to PLOS Mental Health. After careful consideration, we feel that it has merit but does not fully meet PLOS Mental Health’s publication criteria as it currently stands. Therefore, we invite you to submit a revised version of the manuscript that addresses the points raised during the review process.

We look forward to receiving your revised manuscript.

Kind regards,

Renato de Filippis, MD, PhD

Academic Editor

PLOS Mental Health

Journal Requirements:

https://journals.plos.org/mentalhealth/s/figures

https://journals.plos.org/mentalhealth/s/figures#loc-file-requirements

2. We have amended your Competing Interest statement to comply with journal style. We kindly ask that you double check the statement and let us know if anything is incorrect.

3. We noticed that you used “unpublished data" in the manuscript. We do not allow these references, as the PLOS data access policy requires that all data be either published with the manuscript or made available in a publicly accessible database. Please amend the supplementary material to include the referenced data or remove the references.

4. In the online submission form, you indicated that [The data that support the findings of this study are available on request from the corresponding author.].

a. In a public repository,

b. Within the manuscript itself, or

c. Uploaded as supplementary information.

Additional Editor Comments (if provided):

The reviewers identified some valuable points, but also several limitations that need to be addressed. I encourage the authors to address all concerns raised by the reviewers point by point. Thank you

Reviewers' comments:

Reviewer's Responses to Questions

**Comments to the Author**

1. Does this manuscript meet PLOS Mental Health’s publication criteria?

Reviewer #1: Yes

Reviewer #2: Yes

Reviewer #3: Yes

2. Has the statistical analysis been performed appropriately and rigorously?

Reviewer #1: No

Reviewer #2: No

Reviewer #3: Yes

3. Have the authors made all data underlying the findings in their manuscript fully available (please refer to the Data Availability Statement at the start of the manuscript PDF file)?

Reviewer #1: No

Reviewer #2: Yes

Reviewer #3: Yes

4. Is the manuscript presented in an intelligible fashion and written in standard English?

Reviewer #1: Yes

Reviewer #2: Yes

Reviewer #3: Yes

Reviewer #1: 1. The manuscript addresses an important clinical question, but the central research objective could be more explicitly stated in the Introduction.

2. Organization is generally logical, though the dense results table would benefit from being split by diagnostic category for readability.

3. There is excessive repetition between the Results and Discussion sections; condensing the Results and deepening the analysis is advised.

4. Mechanisms of action are well summarized, but a visual schematic would enhance reader understanding.

5. Differences between IR and XL formulations are mentioned but require clearer comparative analysis, particularly regarding tolerability.

6. The review would be strengthened by a more detailed explanation of the heterogeneity in populations and its implications for generalizability.

7. References are comprehensive, but a follow-up on clinical trials marked as “unpublished” may reveal newer published data.

8. Case reports use the term “remission” ambiguously—please clarify the criteria used for this designation.

9. Language is generally professional, though occasionally leans into overinterpretation; be cautious when inferring causality.

10. The authors assert that doxazosin may be more tolerable than prazosin but provide limited comparative data to support this.

Reviewer #2: General comments:

The paper is adding information on the alternative therapy of using Doxazosin in the management of mental health disorders, which are increasingly a serious problem worldwide. Doxazosin has been reported in the treatment of various mental illnesses including PTSD and PTSD-related nightmares. However, the paper is addressing its role in other mental illnesses. However, following are some comments that need to be addressed.

The authors need to work on the grammar and the use of tenses throughout the paper.

The authors need to clearly differentiate the noradrenergic system pathway in the brain and the sympathetic nervous system (SNS) is a division of the autonomic nervous system and a component of the peripheral nervous system, which is responsible for the body's "fight-or-flight" response. The derangement in the noradrenergic system pathways in the brain is responsible for development of mental disorders especially the depression, anxiety and many others. Therefore the authors need to clearly distinguish this. The noradrenergic system pathway has various receptors located in the brain and SNS periphery that may be targeted by drugs like alpha 1 (α1) antagonists such as doxazosin, prazosin etc., that may also cross the BBB to influence the brain and hence their use in mental illnesses.

1. Abstract

The authors should provide a background on the topic addressed in the paper. The aim of the review or research question addressed in the paper is not clear and it needs to be re-written. The statement in line 34-35, “--- included all types of human research, including interventional and observational clinical trials, case series/reports, and unpublished clinical trials”. How were the unpublished clinical trials retrieved, and this also falls in the gray literature? The study also seem not to have followed the “PICO (or PICOS) since a number of papers with different study designs were used in the review. Can the authors clarify this? Then, the papers that were not in PubMed, Embase, Cochrane, and PsycINFO were not considered and this possibly created the selection bias, how was this handled? Couldn’t the use of search engines also provide additional information on the subject? The authors should clearly state who exactly did the final screening of the papers to be included in the final review process. The conclusion in the abstract need to be paraphrased based on the key findings of the review paper.

2. Introduction

Line 57, the authors should include the role of the HPA (Hypothalamic-Pituitary-Adrenal) axis in the modulation of the body's response to stress which in turn may influence mental health. The aim of the review or research question addressed in the paper is not clear and it needs to be re-written (see above comment in the abstract). Specifically, which mental disorders where targeted by doxazosin and therefore used in the review?

3. Methods

The study also seem not to have followed the “PICO (or PICOS) since a number of papers with different study designs were used in the review. Can the authors clarify this? Then, the papers that were not in PubMed, Embase, Cochrane, and PsycINFO were not considered and this possibly created the selection bias, how was this handled? Couldn’t the use of search engines also provide additional information on the subject? The authors should clearly state who exactly did the final screening of the papers to be included in the final review process. The conclusion in the abstract need to be paraphrased based on the key findings of the review paper (see above comments in the abstract). The authors could also have used the PICO or PICOS in the eligibility criteria especially in the inclusion. The authors should also include a section on how included papers were retrieved, screened and who made the final screening of the papers that were used in the review.

4. Results

The authors should cite the table and figure in the text. Also, because of lack of use of PICO or PICOS in the study design, the authors couldn’t aggregate the findings. Suggestion, if possible; can the authors’ further aggregate the finding (outcomes) of those papers with similar study design?

5. Discussion

The authors should also highlight the key limitations of the review.

5. Conclusion

The authors should highlight the key findings of the review with the title and objective if the review as guide.

Reviewer #3: Strengths

Timely and relevant topic: Off-label use of doxazosin in psychiatry is underexplored, and this review compiles dispersed literature.

Comprehensive literature search: Databases (PubMed, Embase, Cochrane, PsycINFO) were systematically searched following PRISMA-ScR.

Clear categorization: The review differentiates between PTSD/nightmares, substance use disorders, and dual diagnoses.

Consideration of pharmacogenetics: Highlights genotype-specific response patterns in cocaine use disorder.

Clinical relevance: The paper identifies gaps in current practice and provides a foundation for future trials.

Major Weaknesses & Recommendations

Methodological Limitations

No risk-of-bias assessment was performed for included studies. While this is a scoping review, a brief discussion of study quality would strengthen the interpretation of findings.

Many included studies are case reports or small open-label trials, yet this limitation is not sufficiently emphasized in the abstract and conclusion.

Presentation & Structure

Tables are detailed but dense. Condensed summary tables (e.g., separating RCTs from case reports) would improve readability.

Figures such as a PRISMA flow diagram should be included to visualize study selection.

Interpretation of Findings

Conclusions could be more cautious regarding clinical recommendations, emphasizing the limited RCT evidence.

The manuscript should explicitly discuss differences between doxazosin IR and XL formulations, as tolerability and dosing appear critical.

References and Consistency

Some references are repeated (e.g., De Jong et al., 2010b is cited twice). Ensure reference style is consistent with PLOS guidelines.

Language & Style

Minor grammatical and typographical issues should be revised for clarity and conciseness.

The abstract could more clearly reflect the limited evidence and exploratory nature of the review.

**Do you want your identity to be public for this peer review?** For information about this choice, including consent withdrawal, please see our Privacy Policy

Reviewer #1: **Yes: ** Bala Nimmana

Reviewer #2: No

Reviewer #3: **Yes: ** Yohanes Sime Tola

---

## [Editor Report · Decision Letter 1]

17 Oct 2025

PMEN-D-25-00312R1

Doxazosin for the treatment of mental health disorders: A scoping review

PLOS Mental Health

Dear Dr. Richardon,

Thank you for submitting your manuscript to PLOS Mental Health. After careful consideration, we feel that it has merit but does not fully meet PLOS Mental Health’s publication criteria as it currently stands. Therefore, we invite you to submit a revised version of the manuscript that addresses the points raised during the review process.

We look forward to receiving your revised manuscript.

Kind regards,

Renato de Filippis, MD, PhD

Academic Editor

PLOS Mental Health

Journal Requirements:

Additional Editor Comments (if provided):

I thank the authors for their submission and patience, and I invite them to work to improve the manuscript as requested by the reviewers, responding point by point.
---

## [Editor Report · Decision Letter 2]

4 Nov 2025

Doxazosin for the treatment of mental health disorders: A scoping review

PMEN-D-25-00312R2

Dear Richardon,

We are pleased to inform you that your manuscript 'Doxazosin for the treatment of mental health disorders: A scoping review' has been provisionally accepted for publication in PLOS Mental Health.

Best regards,

Renato de Filippis, MD, PhD

Academic Editor

PLOS Mental Health

I thank the authors for their work in improving the manuscript.